# AI-ASSISTED AUTHORING FOR TRANSPARENT, DATA-DRIVEN DOCUMENTS

## ABSTRACT

We introduce *transparent documents*, interactive web-based scholarly articles which allow readers to explore the relationship to the underlying data by hovering over fragments of text, and present an LLM-based tool for authoring transparent documents, building on recent developments in data provenance for general-purpose programming languages. As a target platform, our implementation uses Fluid, an open source programming language with a provenance-tracking runtime. Our agent-based tool supports a human author during the creation of transparent documents, identifying fragments of text which can be computed from data, such as numerical values selected from records or computed by aggregations like sum and mean, comparatives and superlatives like "better than" and "largest", trend-adjectives like "growing", and similar quantitative or semi-quantitative phrases, and then attempts to synthesise a suitable Fluid query over the data which generates the target string. The resulting expression is inserted into the article's web page, turning the static text fragment into an interactable data-driven element able to reveal the data that underwrites the natural language claim. We evaluate our approach on a subset of SciGen, an open source dataset consisting of tables from scientific articles and their corresponding descriptions, which we extend with hand-generated counterfactual test cases to evaluate how well machine-generated expressions generalise. Our results show that gpt4o is often able to synthesise compound expressions extensionally compatible with our gold solutions.

## 1 INTRODUCTION: TRANSPARENT, DATA-DRIVEN DOCUMENTS

When interpreting or verifying data-driven claims, a key challenge lies in tracing specific claims back to the relevant data. In peer review, for example, empirical claims typically lack author-supplied links to data, making them hard for reviewers to check directly (Weber & Karcher, 2020). Paper retractions, meanwhile, are often attributable not to fraud, but to simple errors in data management or analysis (Hu et al., 2025). The use of large language models (LLMs) to interpret scholarly documents has seen considerable attention recently, from fact-checking (Abu Ahmad et al., 2025) to interpretation of charts and figures (Roberts et al., 2024), but current LLM interfaces do not support direct interrogation of visual or other outputs for traceability to inputs.

Recent advances in data provenance and data visualisation (Psallidas & Wu, 2018; Bond et al., 2025), on the other hand, have pushed in this direction using a more infrastructural approach. These approaches link computed outputs to their data sources directly by tracking dependency information. This allows visual outputs to support *provenance queries*, user interactions (e.g. mousing over visual elements) that reveal how output features relate to data. The advantage of this approach is that the relationships to data sources are exposed automatically via trusted infrastructure, typically a query language or general-purpose programming language which tracks how data flows through a computation. However, these approaches are limited to outputs computed from data, such as visualisations. What is missing is a way to extend these "direct interrogation" features to natural language itself, where the main claims of most scholarly articles are actually made.

In this paper, we address this gap by combining two complementary approaches: the ability of LLMs to understand technical language and synthesise queries over data, plus the provenance-tracking infrastructure of an open source programming language called Fluid (https://f.luid.org/) (Perera

tableData (15 of 15)

| acc | model | param | time_s |
|---|---|---|---|
| 1 | 80.72 LSTM | 5977 | 99 |
| 2 | 81.73 BiLSTM | 7059 | 106 |
| 3 | 81.97 2 stacked BiLSTM | 9221 | 207 |
| 4 | 81.53 3 stacked BiLSTM | 11383 | 310 |
| 5 | 81.37 4 stacked BiLSTM | 13546 | 411 |
| 6 | 82.64 S-LSTM | 8768 | 65 |
| 7 | 80.35 CNN | 5637 | 34 |
| 8 | 80.97 2 stacked CNN | 5717 | 40 |
| 9 | 81.46 3 stacked CNN | 5808 | 47 |
| 10 | 81.39 4 stacked CNN | 5855 | 51 |
| 11 | 81.03 Transformer (N=6) | 7234 | 138 |
| 12 | 81.86 Transformer (N=8) | 7615 | 174 |
| 13 | 81.63 Transformer (N=10) | 8004 | 214 |
| 14 | 82.37 BiLSTM+Attention | 7419 | 126 |
| 15 | 83.07 S-LSTM+Attention | 8858 | 87 |

As shown in Table 3, BiLSTM gives significantly better accuracies compared to uni-directional LSTM2, with the training time per epoch growing from 99 seconds to 106 seconds. Stacking 2 layers of BiLSTM gives further improvements to development results, with a larger time of 207 seconds. 3 layers of stacked BiLSTM does not further improve the results. In contrast, S-LSTM gives a development result of 82.64 %, which is significantly better compared to 2-layer stacked BiLSTM, with a smaller number of model parameters and a shorter time of 65 seconds. We additionally make comparisons with stacked CNNs and hierarchical attention (Vaswani et al., 2017), shown in Table 3 (the CNN and Transformer rows), CNN is the most efficient among all models compared, with the smallest model size. On the other hand, a 3-layer stacked CNN gives an accuracy of 81.46 %, which is also the lowest compared with BiLSTM, hierarchical attention and S-LSTM. The best performance of hierarchical attention is obtained by S-LSTM+Attention in terms of both accuracy and efficiency. S-LSTM gives significantly better accuracies compared with both CNN and hierarchical attention. Table 3 additionally shows the results of BiLSTM and S-LSTM when external attention is used Attention leads to improved accuracies for both BiLSTM and S-LSTM in classification, with S-LSTM still outperforming BiLSTM significantly.

tableData (15 of 15)

| acc | model | param | time_s |
|---|---|---|---|
| 1 | 80.72 LSTM | 5977 | 99 |
| 2 | 81.73 BiLSTM | 7059 | 106 |
| 3 | 81.97 2 stacked BiLSTM | 9221 | 207 |
| 4 | 81.74 3 stacked BiLSTM | 11383 | 310 |

As shown in Table 3, BiLSTM gives significantly better accuracies compared to uni-directional LSTM2, with the training time per epoch growing from 99 seconds to 106 seconds. Stacking 2 layers of BiLSTM gives further improvements to development results, with a larger time of 207 seconds. 3 layers of stacked BiLSTM further improves the results. In contrast, S-LSTM gives a development result of 82.64 %, which is significantly better compared to 2-layer stacked BiLSTM, with a smaller number of model

Figure 1: Two versions of a transparent document, showing text fragments linked to data

et al., 2022; Bond et al., 2025). Together, these two technologies enable the creation of *transparent documents*, web-based scholarly articles with two key transparency features:

1. **Data-driven:** Quantitative statements expressed in natural language — e.g. that system $X$ is faster than system $Y$ on some task — are computed from the relevant data, rather than occurring merely as static fragments of text.

2. **Data linking:** Readers and reviewers can interactively trace such claims back to the specific data elements that support them, through embedded provenance queries.

Figure 1, generated from our implementation, illustrates these two features. The upper section shows a "transparent" excerpt from Zhang et al. (2018), a scholarly article comparing text encoding techniques. When a reader hovers over the phrase "does not further improve", the relevant data are highlighted on the left. Other fragments (e.g. "better than", "further improvements") that refer to the same data are also marked, allowing the reader to explore supporting and contrasting evidence. The lower section shows a counterfactual situation where the authors' experiments had produced different results: here the phrase "does not further improve" is replaced by "further improves".

This transparent version of the document was implemented in Fluid. The source code is shown in Figure 2, and makes use of several helper functions, a representative subset of which are shown in Figure 5. What makes our solution interesting is that the provenance-tracking runtime of Fluid *and* the LLM-based authoring support are both essential components of the solution, with Fluid providing the interactions, and the LLM-based tool making the authoring process feasible. Generating code for a traditional language like Python would still result in a data-driven document, but crucially without the interactive provenance queries; and without AI-based tooling to support the authoring process, the author would be faced with creating the code in Figure 2 by hand, which is unlikely to be feasible as part of the usual scientific writing process.

AI-assisted authoring of transparent documents thus support turning static text into interactable, data-driven content able to expose the evidential basis of scholarly claims. We envisage two use cases. First, when **authoring** content for an online article, a journalist or scientific publisher may wish to provide text which is linked to the underlying data so that the evidence base for the textual claims can be explored directly from the article. Second, when **reading** a document reporting on findings derived from open data (perhaps a scientific paper or climate report), the reader may want to retroactively interpret parts of the text as queries over the available data and gradually "rationally reconstruct" the relationship between claims in the paper and the evidence base. This might be just to aid their own comprehension, or part of a formal peer review process.

```
let model_ name = model name tableData in f"""
  As shown in Table 3, BiLSTM gives significantly
  ${trendWord (model_ "BiLSTM").acc (model_ "LSTM").acc betterWorse}
  accuracies compared to uni-directional LSTM2, with the training time per epoch
  ${trendWord (model_ "BiLSTM").time_s (model_ "LSTM").time_s growShrink} from
  ${(model_ "LSTM").time_s} seconds to ${(model_ "BiLSTM").time_s} seconds.
  ...
  We additionally make comparisons with stacked CNNs and hierarchical attention (Vaswani et al., 2017),
  shown in Table 3 (the CNN and Transformer rows),
  ${(findWithKey_ "time_s" (minimum (map_ (fun y → y.time_s) tableData)) tableData).model} is the
  ${rankLabel "most efficient" (findIndex "model" "CNN" (sort (fun a b → a.time_s < b.time_s) tableData))}
  among all models compared, with the
  ${rankLabel "smallest" (findIndex "model" "CNN" (sort (fun a b → a.param < b.param) tableData))}
  model size. On the other hand, a 3-layer stacked CNN gives an accuracy of
  ${(model "3 stacked CNN" tableData).acc} %, which is also the
  ${rankLabel "lowest" (findIndex "model" "CNN" (sort (fun a b → a.time_s < b.time_s) tableData))}
  compared with BiLSTM, hierarchical attention and S-LSTM. The
  ${rankLabel "best" (findIndex "model" "S-LSTM+Attention" (sort (fun a b → b.acc < a.acc) tableData))}
  performance of hierarchical attention is obtained by S-LSTM+Attention in terms of both accuracy
  and efficiency. S-LSTM gives significantly
  ${trendWord (model_ "S-LSTM").acc (model_ "CNN").acc betterWorse}
  accuracies compared with both CNN and hierarchical attention. Table 3 additionally shows the results of
  BiLSTM and S-LSTM when external attention is used. Attention leads to improved accuracies for both
  BiLSTM and S-LSTM in classification, with S-LSTM still
  ${trendWord (model_ "S-LSTM").acc (model_ "BiLSTM").acc underOverPerforming} BiLSTM significantly.
"""
```

Figure 2: Gold solution for transparent document in Figure 1 (some lines omitted)

**Contributions.** Our specific contributions are as follows. We leave implementing a full Copilot-like authoring plugin for an IDE such as VSCode or Cursor for future work (Section 6).

- A proof-of-concept LLM-based tool for iteratively transforming a preexisting opaque document and associated data set into a transparent, data-driven counterpart (Section 2);
- A summary of the natural language idioms we have studied (Section 3) and an empirical evaluation of how well state-of-the-art models are able to solve the associated interpretation and code synthesis problems (Section 4).

## 2  AI-ASSISTED AUTHORING WORKFLOW

Our authoring tool is composed of two LLM-based agents. A **SuggestionAgent** identifies text fragments potentially computable from data, and an **InterpretationAgent**, given a text fragment provided by the SuggestionAgent or by the author, attempts to synthesise a Fluid expression which computes the target fragment. The main components of the workflow are as follows:

1. **Initial configuration.** The author imports the target text and accompanying data into the system to create a programmatic representation of the target document. Initially this is simply equivalent to the target text, taking the form of a string literal """..."""", where the triple quotes are Fluid syntax for a Python or JavaScript-style *interpolated string*, i.e. a literal where expressions of the form $\{e\}$ are permitted within the string. The SuggestionAgent analyses the target text and identifies any fragments which are candidates for being computed instead of remaining as literal substrings.

2. **High-level Authoring workflow.** The system then enters the human-in-the loop authoring workflow shown in Figure 3, where the author interacts with the InterpretationAgent. The system waits for the author to select a fragment of text $s$ to interpret (perhaps previously highlighted by the SuggestionAgent). The system then attempts to generate a candidate Fluid expression $e$ using the closed-loop synthesis step (3) below. If code synthesis succeeds with an expression $e$, the system proceeds to the manual validation step (4) below. If the synthesis step fails with no expression, no remedial action is possible; this is considered an unsuccessful path through the workflow and returns the system to the entry state. Otherwise the synthesis step produces an expression $e$ which evaluates to a mismatched string $s' \neq s$ outcome, and the user can choose to manually abort and return to the entry state, or optionally to *revise the goal*, replacing $s$ with $s'$

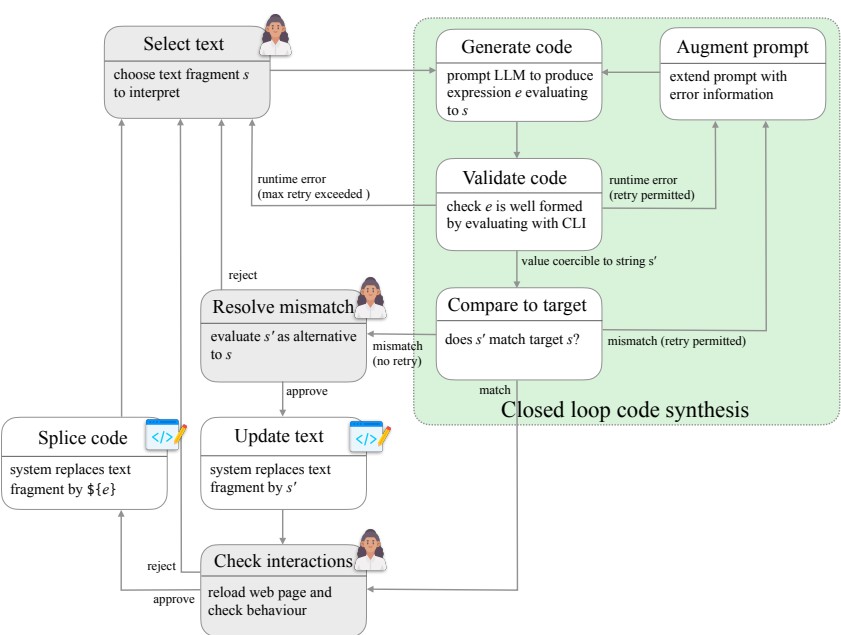

Figure 3: Human-in-the-Loop workflow (states requiring human intervention in grey)

in the target document and retaining $e$ as the candidate expression. This is intended to cover the situation where the author has made a claim which is *incorrect*, and the data set and surrounding natural language have led the LLM to synthesise an expression which generates a different value from the one specified by the user.

3. **Code synthesis step.** The expression synthesis step is an error-guided iterative prompting loop (Skreta et al., 2023), beginning with an initial prompt sent to the LLM (see *Prompt design* below) requesting the generation of an expression $e$. Using the Fluid command-line interface, the expression is validated to check that it evaluates without error, produces a value coercible to a string $s'$, and finally that $s'$ is equal to the target fragment $s$. Any failure triggers prompt augmentation with the appropriate error message and the system retries generation. If code synthesis loop is able to yield an expression which computes $s$ within a maximum number of retries, the synthesis step succeeds with $e$. If the last generated $e$ was invalid (resulting in an error), the code synthesis step fails with no expression. Otherwise, code synthesis produces an expression $e$ but with a mismatched string outcome $s' \neq s$.

4. **Manual validation step.** Once a candidate expression has been generated, the system replaces the selected substring $s$ with the interpolation expression $\{e\}$, creating a new (but only tentative) document configuration. The author can republish the web page hosting the document and interact with the proposed revision. As shown in Section 4, this is an important validation step that can reveal errors in the generated expression. If the interactions look reasonable, the author can approve the new document state; this is the primary successful path through the workflow and returns the system to the entry state where it is waiting for another top-level interaction from the author. Otherwise, the author rejects the proposed change and returns to the entry state without any change to the document.

This human-in-the-loop design combines automated synthesis with validation and author oversight, providing a substantial level of automation, but requiring the author to intervene at key steps to ensure correctness.

**InterpretationAgent prompt design.** The InterpretationAgent is guided by a structured system prompt that frames code generation as a precise replacement task. The model receives the imported datasets, helper modules, and the current Fluid representation of the paragraph, in which a text fragment is marked with the tag $[\text{REPLACE} \dots]$. The task is to substitute this placeholder with a Fluid expression that evaluates exactly to the target string, reconstructing quantitative or comparative

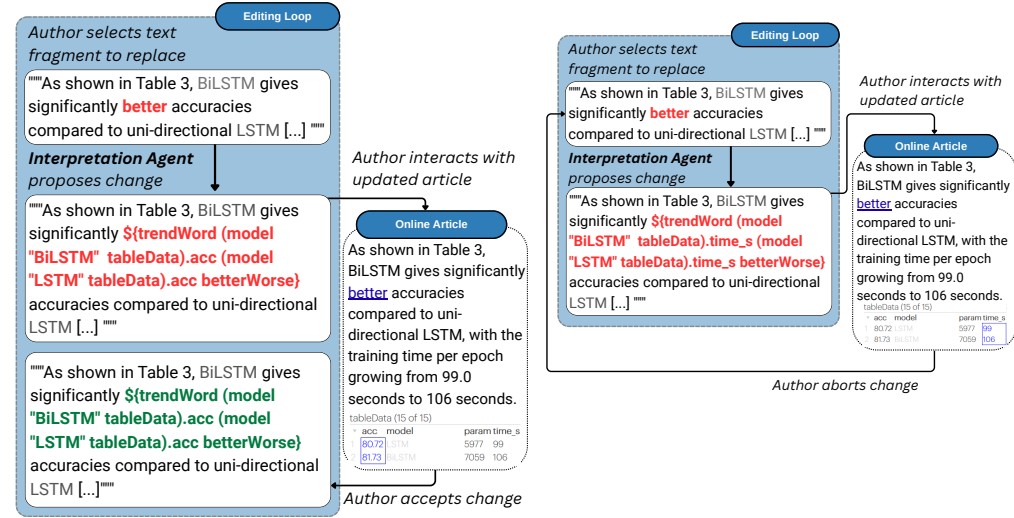

(a) Author accepts expression          (b) Author identifies error and rejects

Figure 4: Two possible paths through editing loop, with interactive verification of generated code

claims as data queries. To ensure integration with the workflow, the output must consist solely of a syntactically valid Fluid expression, with no additional commentary. The full prompt is given in Appendix A.

## 3 TARGET IDIOMS OF NATURAL LANGUAGE

| Label | Example | Gold Solution for Example |
|---|---|---|
| Data retrieval | the training time per epoch growing from 67 seconds to 106 seconds. | `(model_ "LSTM").time_s` |
| Ratio | The Energy Sector accounts for total methane emissions of 52.80% in 2030. | `(getByCategory "Energy Sector" year).emissions /`
`  sum (map (fun x -> x.emissions)`
`    (getByYear year)) * 100` |
| Average | The average methane emissions for the year 2030 is 13.51 | `sum (map (fun x -> x.emissions)`
`  (getByYear year)) / length records` |
| Min/Max | The Energy Sector recorded its highest methane emissions in 2030 | `let maxEntry = maximumBy (fun x -> x.emissions)`
`    (filter (fun x -> x.type == "Energy Sector")`
`      tableData)`
`in maxEntry.year` |
| Rank | 3-layer stacked CNN gives an accuracy of 81.46%, which is the lowest compared with BiLSTM, and S-LSTM | `rankLabel "lowest"`
`  (findIndex "model" "CNN"`
`    (sort cmpTime tableData))` |
| Sum | The total methane emissions for the year 2030 is 37.74 for Agriculture | `sum (map (fun x -> x.emissions)`
`  (getByYear year))` |
| Comparison | The training time per epoch growing from 67 seconds to 106 seconds. | `trendWord`
`  (model_ "BiLSTM" tableData).time_s`
`  (model_ "LSTM" tableData).time_s`
`  growShrink` |
| Generalised quantifiers | In the case of one syndrome (Hemorrhagic) we noticed an unusually low level of recall for SVM but not for NB. | `unusuallyHighLow (overallComparison [`
`  compareCols col "naive_bayes_r"`
`    (findWithKey_ "synd" "Hem." tableData)`
`  | col <- ["svm1_r", "svm2_r", "svm3_r", "svmr_r"]`
`])` |

Table 1: Quantitative/semi-quantitative natural language forms considered in this paper

```
1  let ordinalMap =                                    19  let trendWord n1 n2 compareWord =
2    [ { lastDigit: 1, suffix: "st" },                 20      compareWord (compare n1 n2);
3      { lastDigit: 2, suffix: "nd" },                 21
4      { lastDigit: 3, suffix: "rd" } ];               22  let growShrink EQ = "unchanging";
5                                                      23      growShrink LT = "shrinking";
6  let ordinal n =                                      24      growShrink GT = "growing";
7    if n <= 0 then error "n <= 0 not supported"        25
8    else if (n < 4) then                               26  let smallerHigher EQ = "equal";
9      numToStr n ++                                    27      smallerHigher LT = "smaller";
10     (findWithKey_ "lastDigit" n ordinalMap).suffix   28      smallerHigher GT = "larger";
11   else if (n >= 4) 'and' (n <= 20) then              29
12     numToStr n ++ "th"                               30  let improvements EQ = "no further improvements";
13   else error "n > 20 not supported";                 31      improvements LT = "no further improvements";
14                                                      32      improvements GT = "further improvements";
15 let rankLabel word n =
16   (if n == 1 then "" else ordinal n ++ "-") ++ word;
```

Figure 5: SciGen helper functions (representative examples)

Table 1 summarises the natural language idioms studied in this paper. With state-of-the-art models like *gpt-4o* and *gpt-5*, our system is able to resolve basic table lookups of direct numerical values, as well as computations of percentages, averages, minima and maxima, and totals, each mapped to the corresponding aggregation over the source data. For example, phrases such as "the Energy Sector accounts for 52.80% of total emissions" and "average methane emissions for 2030 is 13.51" are interpreted in terms of sum and mean respectively over the relevant data values. Similarly, "recorded its highest emissions in 2030" is interpreted as a maximumBy query, while a statement such as "CNN gives the lowest accuracy" is mapped to an explicit computation of rank.

We also consider *trend* expressions, which comparative natural language phrases describing how a data attribute evolves over time, such as "training time growing from 67 to 106 seconds". Such idioms are mapped to higher-order functions like trendWord parameterised on additional helper functions such as growShrink and betterWorse (shown in Figure 5) which map comparisons to appropriate natural language phrases.

Taken together, these categories cover a representative portion of the numerical reasoning idioms found in the SciGen benchmark. However, some linguistic forms that commonly arise in scholarly articles are not covered in our analysis. We have yet to study approximate quantitative terms like "around 50%" or "roughly 100 instances", nor interval-based descriptions such as "between 30 and 40%" or "within 5–10 seconds". While we have no reason for thinking these will present specific difficulties, other forms are likely to be more challenging. So-called *graded* modal adverbs (Lassiter, 2017) which modify adjectival comparatives like "better" – as in "slightly better" and "significantly higher" – especially when combined with trends over time, as in "steadily increasing" or "sharply declining" – are likely to prove difficult because the interpretation of these qualifiers can be subjective and context-dependent. Generalised quantifiers like "generally" and "usually" (Barwise & Cooper, 1981) present similar challenges because colloquial use may differ from more formal uses (in some situations "most" might mean a majority, i.e. greater than 50% of cases, but in others may mean only "greater than any other alternative proportion"). On the other hand these difficulties also present themselves to human readers, so extending coverage to these idioms would substantially deepen our tool's ability to bridge natural language reporting with interpretation in terms of the underlying dataset, perhaps revealing inconsistent use of technical language on the part of the author. We discuss this further in Section 6.

## 4    EXPERIMENTAL EVALUATION

### 4.1    RESEARCH QUESTIONS

Our evaluation tests the ability of the InterpretationAgent to translate quantitative and semi-quantitative expressions from scholarly natural language into executable queries that operate on the underlying dataset. Beyond raw accuracy, we are also concerned with how performance varies

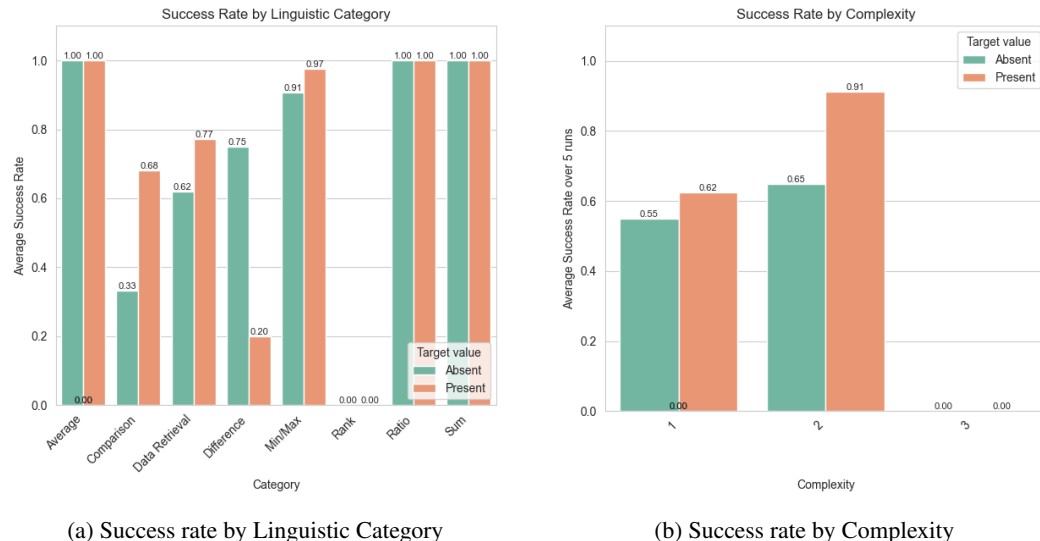

(a) Success rate by Linguistic Category

(b) Success rate by Complexity

Figure 6: Success rate of the proposed system, measured over 5 runs with *gpt-4o*

with task complexity, and whether the generated expressions are robust under changes to data or in the presence of ambiguity or other low data quality issues. These are captured in two research questions:

**RQ1. Interpretation Accuracy across Linguistic Idioms and Complexity.** To what extent can LLMs accurately interpret quantitative and semi-quantitative claims in scholarly text as data queries? We examine performance across a range of linguistic idioms (e.g. averages, percentages, min/max, ranks, as summarised in Table 1) and investigate how accuracy varies with task complexity, measured (somewhat crudely) by the number of query sub-expressions (e.g. retrieval, aggregation, or arithmetic) present in the gold solution.

**RQ2. Generalisability and Robustness.** How well do the generated expressions generalise when the underlying data changes, or when the input contains misleading or ill-specified information? We test whether generated queries continue to produce correct outputs under a set of hand-generated counterfactual modifications of the dataset, based on expected query results specific to each test case, and also how counterfactual performance is impacted by the presence of misleading or adversarial phrasing. Table 2 shows some of cases we deem problematic in this sense; in these cases, producing a valid expression is likely to be challenging because of ambiguities in the input data or accompanying natural language.

### 4.2 RESULTS

**Interpretation Accuracy across Linguistic Idioms and Complexity.** To evaluate RQ1, we used a sample of the SciGen dataset (Moosavi et al., 2021), an open source dataset consisting of tables from scientific articles and their corresponding descriptions. We aggregated the results according to the linguistic categories from Table 1. Figure 6a illustrates the success rate for each category, both with and without target-value sharing.

The results show that *the system is robust when provided with sufficient guidance but degrades when underspecified.* With the target-value sharing, the InterpretationAgent produced correct Fluid expressions in 74.9% (S.D. 3.0%) of cases, but performance dropped to 57.1% when the target was withheld. This highlights the system's reliance on explicit cues when resolving ambiguous fragments.

*Performance also varied across linguistic categories.* Success rates exceeded 68% for comparison, 77.3% for data retrieval, and 97% for min/max search tasks. In contrast, accuracy decreased significantly for expressions requiring differences (20%) and for ranking tasks (0%).

| Problem Type | Example | Explanation |
|---|---|---|
| false comparison | BiLSTM is the most efficient among all models compared, with the highest model size | BiLSTM is not the most efficient, nor does it have the largest size. |
| wrong numerical value | LSTM is the fastest model with overall time taken being 90 seconds | It is not 90 but 106. |
| ambiguous referent | LSTM is the fastest model with overall time taken being 90 seconds | There are two type of time in the dataset (training_time, execution_time), both with a value of 90 seconds. |

Table 2: Categories of problematic example

The trend for compositional complexity is more nuanced as shown in Figure 6b, which reports the success rate as a function of the number of categories assigned to each expression: success rates are 62% for single-category expressions, increase to 91% when two categories are combined, but collapse to 0% when three categories are involved. This suggests that *moderate composition can actually aid performance, perhaps by giving the model clearer structural cues, but that complexity beyond a certain threshold overwhelms the synthesis process.*

**Generalisability and Robustness.** As a preliminary attempt to address RQ2, we carried out *counterfactual testing* to evaluate the robustness of generated expressions under changes to the underlying data. In this setup, the input tables were modified according to hand-craft test specifications, and both the expected and generated expressions were re-executed to check whether the behaviours remained consistent. Across 300 test executions, 121 contained at least one counterfactual error (an average of 3.8 per case), of which 42 ultimately still succeeded. These tests highlight cases where an expression may coincidentally yield the correct output on the original data but fails to be extensionally equivalent more generally (i.e. under perturbation). For example, in one test the system generated

$$(\text{findWithKey}\_ \text{"model"} \text{"LSTM2"} \text{tableData}).\text{time}\_\text{s}$$

intended to retrieve the execution time of the LSTM model, but incorrectly referred to LSTM2. Counterfactual testing exposed this mismatch, which would otherwise have gone undetected.

At present, counterfactual tests are used only as an evaluation device, not as part of the authoring workflow itself. For future work (Section 6), we plan to investigate automatic generation of counterfactual tests, allowing these additional robustness checks to be integrated into the document authoring workflow.

## 5 RELATED WORK

**Argument mining.** *Argument mining* is an area of NLP which involves identifying argumentative structures in text, such as claims, premises, and conclusions, and mapping them to formal representations (Palau & Moens, 2009; Lippi & Torroni, 2015). Early work focused on rule-based approaches, while more recent work has leveraged machine learning and deep learning techniques (Stab & Gurevych, 2014b; 2017; Eger et al., 2017). The field has also emphasized defining annotation schemes for the task, such as the Argumentative Zoning framework (Teufel & Moens, 2002; Teufel et al., 2009), as well as schemes more directly tailored to argument mining (Stab & Gurevych, 2014a). The field has focused on various domains, starting from legal texts (Toulmin, 2003), and has relied on online resources such as Debatepedia (Cabrio & Villata, 2013). The community has also rapidly engaged with work that explores the use of argument mining in scientific texts (Liakata et al., 2012; Lauscher et al., 2018b) to better understand the structure of scientific arguments and the relationships between different claims and evidence. While the advent of LLMs has improved performance (Gorur et al., 2025; Vrakatseli et al., 2025), argument mining remains a challenging task, particularly when it comes to identifying implicit argumentative relations between discourse units, and reasoning about relationships among different argumentative components, especially in cross-domain settings where models struggle to generalise (Gemechu et al., 2024). While in our work we

do not directly perform traditional argument mining, this work similarly relies on the identification of claims in text, which we evaluate following established practices in the field.

**NLP and scientific writing.** The intersection of NLP and scientific writing has gained increasing attention in the last decade, with a focus on improving the clarity, coherence, and overall quality of scientific texts. On the authoring side, tools such as automated writing assistants can support researchers in producing more fluent and accessible text, for instance through grammar correction, summarisation and text simplification (Napoles et al., 2017; Stiennon et al., 2020; Takeshita et al., 2024; Saggion & Hirst, 2017). Other approaches specifically target the argumentative structure of scientific papers, helping writers to organise contributions and claims more effectively (Lauscher et al., 2018a). Nowadays, general purpose LLMs such as ChatGPT or tools tailored for the task such as Grammarly show the variety of support that NLP tools can provide to authors (Wu et al., 2023; Ahn, 2024; Khalifa & Albadawy, 2024).

At the same time, NLP methods are being developed to assist reviewers and editors in evaluating submissions. These include systems for detecting potential issues such as lack of clarity, weak argumentative support, or even factual inconsistencies and up to scientific fraud (Thakkar et al., 2025; Fromm et al., 2021; Freedman & Toni, 2024). Such tools can also facilitate meta-reviewing by providing summaries of peer reviews and identifying points of disagreement among reviewers (Kumar et al., 2023). While AI tools show promises in improving the peer-review process (Tyser et al., 2024), there are also various risks associated such as breaches of confidentiality, lack of transparency and biases (Perlis et al., 2025). Our current work situates itself in the context of NLP tools for supporting the understanding of scientific writing; specifically, it addresses one of the major critiques toward the automation of such process by offering a transparent way of examining its workflow.

**Interpretable NLP.** As Figure 1 illustrates, scientific texts routinely make use of comparatives like "faster" while leaving one of the argument slots implicit, with the context determining the omitted referent. LLMs demonstrate considerable competence in resolving these and other more syntactic forms of anaphora such as pronouns (Zhu et al., 2025), but the resolved referent itself – concretely, what was being referred to – remains implicit. Interpretable NLP is a recent research direction which aims to support comprehension (and production) of text in a more explicit and transparent way (He, 2023). By generating code that formalises the interpretation of a comparative like "faster", our approach also makes these implicit references explicit; combining our system with interpretable NLP would allow the user to explore the linguistic interpretation as well.

## 6 CONCLUSIONS AND FUTURE WORK

We introduced a proof-of-concept system for authoring transparent, data-driven documents by combining LLM-based code synthesis with Fluid's provenance-tracking runtime. Our evaluation on SciGen shows that the approach can reliably link natural language claims to their underlying data, while also revealing common failure modes such as ambiguity and misleading input.

Future work includes reducing reliance on predefined helper functions such as growShrink and trendWord. While there is an advantage in using a predefined set of helpers (in that they offer a uniform framework for interpreting a given scholarly document), we also aim to enable the system to operate in their absence, for instance by turning "definition not found" errors into augmented prompts that trigger automatic generation of missing definitions. We also plan to broaden the scope of supported artifacts, extending interpretation to visualisations and intermediate datasets derived from cleansing or aggregation, and to cover additional idioms such as cardinals, multiplicatives, rounding, and graded adjectives.

Another priority is improving integration and validation. Embedding the system into developer and authoring environments such as VSCode or Cursor would make the workflow more seamless, while automatic generation of counterfactual test cases could strengthen validation at authoring time. Finally, distinguishing between *referential terms* with fixed denotations and queries with data-dependent values may help in repairing false or inconsistent statements, ensuring that generated expressions remain aligned with both the data and the author's intent.

## REPRODUCIBILITY STATEMENT.

To facilitate reproducibility, we provide a zip archive in the supplementary materials containing the complete source code, the datasets used in our experiments, and a README file with detailed instructions for running the scripts.

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

APPENDICES

# A INTERPRETATIONAGENT SYSTEM PROMPT

You are a specialized language model for the Fluid functional programming language.
Your task is to analyze a JSON object that represents the user's Fluid program and its context,
and to generate the Fluid expression that must replace the [REPLACE value=] placeholder inside
the paragraph.

Input Structure
The JSON input always contains:
-datasets: one or more JSON-like arrays containing the data used by the program
(scenario-related key–value pairs).
-imports: Fluid helper libraries provided by the user's program.
-code: Additional Fluid functions and definitions from the user's program.
-paragraph: A description that includes exactly one [REPLACE ...] tag.
-paragraphValue: The correct final version of the paragraph (ground truth).

Note: imports, code, and datasets are part of the user's Fluid program, not just supporting context.
Your output must be consistent with these definitions.

Task
Identify the [REPLACE ...] tag in paragraph.
If the tag has the value property, generate a Fluid expression that evaluates exactly to that value.
If not, infer the correct value by comparing paragraph, paragraphValue, and (if needed) datasets.
The result must always be a Fluid expression that evaluates to a string.

Output Format
Return only the Fluid expression, nothing else.

Constraints
-Output exactly one valid Fluid expression.
-Ensure it is syntactically correct and consistent with the provided imports and code.

# B SUGGESTIONAGENT SYSTEM PROMPT

You are an expression detector for Fluid language.
Fluid is a functional programming language used to represent structured data queries and comparisons in a
transparent way.

TASK DESCRIPTION

Given a natural language paragraph and a structured dataset, identify and annotate the parts of the
paragraph that can be replaced by a Fluid expression.

You must detect:
- Explicit values (e.g., scores, names, numbers)
- Comparative expressions (e.g., *better than*, *worse*, *higher*, *more than*)
- Superlative or aggregated expressions (e.g., *the best*, *highest*, *maximum*, *top performer*)

FORMAT

Replace each detected expression with:

[REPLACE value=...]

Where 'value' contains the **original text** of the expression (e.g., "91.57", "better", "the best") —
not the rewritten logic or Fluid code.

IMPORTANT RULE

When replacing comparative or superlative expressions (like "better", "worse", "the best", "highest"),
the 'value' **must be the exact original word or phrase** from the paragraph.

Correct:
S-LSTM gives [REPLACE value="the best"] reported results.
BiLSTM performs [REPLACE value="better"] than LSTM.

Incorrect:
S-LSTM gives [REPLACE value="getMaxBy f1 data"] results.
BiLSTM performs [REPLACE value="BiLSTM.acc ¿ LSTM.acc"] than LSTM.

If needed, annotate separate values independently:

Example:
BiLSTM gives [REPLACE value="91.2"]% accuracy, which is [REPLACE value="better"] than LSTM.

---

EXAMPLES

Example Fluid code:

let bestModel = getMaxBy f1 data in bestModel.model

---

INPUT EXAMPLE

Paragraph:
For NER (Table 7), S-LSTM gives an F1-score of 91.57% on the CoNLL test set, which is significantly
better compared with BiLSTMs. Stacking more layers of BiLSTMs leads to slightly better F1-scores
compared with a single-layer BiLSTM. Our BiLSTM results are comparable to the results reported
by Ma and Hovy (2016) and Lample et al. (2016).
In contrast, S-LSTM gives the best reported results under the same settings.
In the second section of Table 7,Yang et al. (2017) obtain an Fscore of 91.26%.

Data:
[
  −model: "BiLSTM", f1: 90.96″,
  −model: "2 stacked BiLSTM", f1: 91.02″,
  −model: "3 stacked BiLSTM", f1: 91.06″,
  −model: "S-LSTM", f1: 91.57″,
  −model: "yang2017transfer", f1: 91.26″
]

---

OUTPUT EXAMPLE

For NER (Table 7), S-LSTM gives an F1-score of [REPLACE value=91.57]% on the CoNLL test set,
which is [REPLACE value="better"] compared with BiLSTMs.
Stacking more layers of BiLSTMs leads to [REPLACE value="better"] F1-scores compared with a single-layer BiLSTM.
Our BiLSTM results are comparable to the results reported by Ma and Hovy (2016) and Lample et al. (2016).
In contrast, S-LSTM gives [REPLACE value="the best"] reported results under the same settings.
In the second section of Table 7, Yang et al. (2017) obtain an Fscore of [REPLACE value=91.26]%.

