# OpenReview forum: "AI-Assisted Authoring for Transparent, Data-Driven Documents"
_ICLR.cc/2026/Conference — Submitted to ICLR 2026_

### Official Review · Reviewer_VRkd · 2025-10-26

**Soundness:** 2
**Presentation:** 2
**Contribution:** 2
**Rating:** 2
**Confidence:** 4

**Summary:**

This work proposes a concept and tool called transparent documents, enabling better reader sensemaking of information underlying texts in the document.

**Strengths:**

+ AI-based assistive tools is an important research direction

**Weaknesses:**

- My greatest confusion is: isn't this supposed to be an HCI paper? Shouldn't some version of this go to CHI/CSCW/UIST/...? To me, there isn't any technical contribution for an ML venue beyond getting the system implemented, but this type of contribution should be very suitable for the HCI venues.

- Since "human" is a really central part in this system, it would be nice to get evaluated by actual human/users too. The evaluation, from an LLM/NLP standpoint, is perhaps not enough. It would be nice to show if the system could do well on some document NLP tasks (like scientific literature?) to begin with.

- Many of the niche details, like Table 1, are perhaps unnecessary for the main paper, when under consideration at an ML venue.

**Questions:**

please see above

---

### Official Review · Reviewer_azjF · 2025-10-29

**Soundness:** 3
**Presentation:** 3
**Contribution:** 3
**Rating:** 6
**Confidence:** 3

**Summary:**

This paper introduces transparent, data-driven documents, a new form of scholarly article that allows readers to directly explore how text claims relate to underlying data. Instead of leaving numerical or comparative statements as static text, these documents embed interactive provenance tracking, enabling readers to hover over phrases and see the exact data that supports them.

**Strengths:**

+ This paper primarily visualizes the data credibility of a research article through the collaborative interaction between two agents, forming a complete interactive system for "data credibility" that transparently reveals the authenticity of the paper's conclusions.
+ The issues addressed by the paper include multiple categories, such as the reliability of data related to proportions, maximum and minimum values, and it even enables transparency analysis for ranking comparisons and generalized quantifiers, demonstrating strong practicality.
+ The supplementary experiments in the paper show that the system's interpretation accuracy is moderate, and the robustness and generalizability of the generated Fluid code are also at a moderate level.

**Weaknesses:**

+ The system presented in this paper is more engineering-oriented, with relatively limited academic research value.
+ The experimental design lacks a comprehensive justification. Without comparisons to other baseline methods, it is difficult to determine whether this system represents the optimal solution (though, as a pioneering work, more in-depth ablation studies could be considered). Overall, it can be regarded as a well-executed engineering paper.

**Questions:**

See above

---

### Official Review · Reviewer_Bezq · 2025-10-30

**Soundness:** 3
**Presentation:** 3
**Contribution:** 2
**Rating:** 4
**Confidence:** 4

**Summary:**

This paper introduces an AI-assisted, human-in-the-loop authoring framework that converts static textual claims in scientific writing into transparent, data-driven documents. The system employs two cooperative LLM agents:
- **SuggestionAgent**, which detects fragments that can be programmatically derived from underlying data;
- **InterpretationAgent**, which translates these fragments into executable *Fluid* expressions that compute the textual values and expose their data provenance.

Authors interact with the system through an iterative loop: they inspect the generated code, approve or reject the changes, and can reload the rendered web page to verify correctness interactively. Evaluation on a subset of the SciGen dataset measures success rates of generated expressions across linguistic idioms and compositional complexity, and a set of manually constructed counterfactual tests probes robustness under data perturbations. Results indicate a 74.9 % success rate with target-value sharing (57.1 % without), high accuracy on simple categories such as min/max and retrieval, but near-zero performance on difference and ranking expressions.

**Strengths:**

- This framing connects two currently active research directions: LLM-based code synthesis and data provenance systems. By automatically translating natural language descriptions into executable queries, the authors aim to make scientific communication more transparent and auditable. The idea is conceptually novel and resonates with the broader goal of improving reproducibility in AI-assisted research writing.

- In terms of clarity and presentation, the paper is exceptionally well structured and easy to follow. The authors take care to describe each module, agent interaction, and validation step in concrete terms. Figures such as the workflow diagram (Fig. 3) and category-based performance plots are clearly labeled and informative. The writing style balances technical detail with readability, making the work accessible to both NLP and HCI audiences.

- The proposed system is also a coherent and technically feasible proof-of-concept, and the implementation in the Fluid runtime, which natively supports provenance tracking, makes the idea concretely demonstrable rather than purely theoretical.

**Weaknesses:**

- The main limitation of this work lies in its restricted and preliminary evaluation. The experiments are conducted on a small subset of the SciGen dataset, without a clear description of selection criteria. Although the paper provides useful category-level statistics, it does not analyze statistical variance beyond reporting standard deviation or assess generalization to unseen writing styles or datasets.

- This paper describing a “manual validation step,” but lacks a true user-centered evaluation. No quantitative data are provided about how often human authors approve or reject system suggestions, how long validation takes, or what types of errors are most common. Without such usability evidence, it is hard to assess whether the framework reduces author workload or simply shifts it from writing to debugging.

**Questions:**

- Can authors report statistics about the manual validation process? For examples, the ratio of accepted to rejected edits, average validation time per fragment, or the most common sources of rejection? Such data would illustrate how practical and scalable the workflow is for real authors.

- The paper states that experiments were conducted on “a subsample of the SciGen dataset” (line 367), but it remains unclear how this subset was chosen. Could the authors provide more details on the selection logic, sampling criteria, and the exact number of examples used in evaluation?

- It seems this framework rely on predefined helper routines such as trendWord or growShrink, which encode semantic logic that the model merely invokes rather than learns. Moreover, the large performance gap between target-value-sharing (74.9%) and no-target (57.1%) suggests potential reliance on implicit answer leakage rather than actual LLM reasoning. Could the authors conduct an ablation study removing or varying these helper components to determine how much of the system’s success derives from the LLM’s own reasoning versus predefined components?

---

### Official Review · Reviewer_AyvP · 2025-11-01

**Soundness:** 3
**Presentation:** 3
**Contribution:** 2
**Rating:** 6
**Confidence:** 3

**Summary:**

The paper presents a system to automatically annotate scientific papers with
interactive elements that allow to link claims to the evidence supporting them.
The system is largely based on LLMs. The authors describe it and evaluate it
empirically.

**Strengths:**

The proposed system addresses an important problem in science, in particular
with increasing publication numbers in many fields. The proof of concept shows
that this could potentially help readers understand scientific papers better and
check claims more easily.

**Weaknesses:**

The proposed system still seems to be in early stages with respect to what can
be verified, which limits its usefulness in practice. First, as far as I
understand, it is limited to single papers and does not allow to check claims a
paper makes about results presented in another paper. This is where a system
like the proposed would be most useful -- while directly linking to evidence in
the same paper is useful, manually checking this information is not nearly as
laborious as checking something in another paper.

Second, the claims that can be checked seem to be quite simple (checks for more
complex claims cannot be generated with LLMs) and thus easy to check manually.
Even then, a human has to verify all generated links, which in at least some
cases are wrong. This begs the question of whether a manual system where only
larger and more important claims are annotated would not be more useful in
practice.

**Questions:**

n/a

---

### Author Response · Authors · 2025-11-21

Thanks to reviewers for your comments, we will post a detailed response and initial revision paper on the 24th (AoE).

---

### Meta-Review · Area_Chair_Tx3j · 2026-01-07

**Summary:**

Reviewers noted the paper's novel concept of AI-assisted transparent documents linking claims to data evidence via LLMs, praising its potential for scientific reproducibility and clear structure.

However, criticisms focused on limited scope (simple claims, single-paper focus), engineering-heavy (lacks ML innovation), insufficient evaluation (no user studies, baselines, or robustness), and misalignment with ML venues (more HCI-suited), resulting in a lean reject.

**Reviewer Concerns:**

None, as no rebuttal provided.

Outstanding: Early-stage limitations (simple claims only, no cross-paper checks); evaluation rigor (small subsample, no human validation/time stats, no baselines/ablations); academic fit (HCI-oriented, limited ML novelty); usability/practicality (no user-centered metrics, potential helper routine reliance).

**Reviewer Scores:**

No change since no rebuttal is provided.

---

### Decision · Program_Chairs · 2026-01-26

Reject